# *Pseudomonas* sp. Strain ADAl3–4 Enhances Aluminum Tolerance in Alfalfa (*Medicago sativa*)

**DOI:** 10.3390/ijms26104919

**Published:** 2025-05-20

**Authors:** Yiming Zhang, Yanjun Ji, Fuxin Liu, Yutong Wang, Chengyi Feng, Zhenzhen Zhou, Zijian Zhang, Long Han, Jinxia Li, Mingyu Wang, Lixin Li

**Affiliations:** Key Laboratory of Saline–Alkali Vegetation Ecology Restoration, Ministry of Education, College of Life Sciences, Northeast Forestry University, Harbin 150040, China; syzdxk@163.com (Y.Z.); jiyanjun1204@163.com (Y.J.); liu15686717190@163.com (F.L.); m18567898663@163.com (Y.W.); f18865221735@163.com (C.F.); zzz_1204@126.com (Z.Z.); zhangzijian@163.com (Z.Z.); hanlharry@163.com (L.H.); lijinxia@nefu.edu.cn (J.L.); wmy19970825@163.com (M.W.)

**Keywords:** aluminum stress tolerance, plant growth-promoting rhizobacteria (PGPR), auxin homeostasis, flavonoid biosynthesis, signal transduction

## Abstract

Aluminum toxicity severely inhibits root elongation and nutrient uptake, causing global agricultural yield losses. Dissolved Al^3+^ are accumulating in plants and subsequently entering food chains via crops and forage plants. Chronic dietary exposure to Al^3+^ poses a risk to human health. In this study, *Pseudomonas* sp. strain ADAl3–4, isolated from plant rhizosphere soil, significantly enhanced plant development and biomass. Phenotypic validation using *Arabidopsis* mutants showed that strain ADAl3–4 regulates plant growth and development under aluminum stress by reprogramming the cell cycle, regulating auxin and ion homeostasis, and enhancing the root absorption of Al^3+^ from the soil. Transcriptomic and biochemical analyses showed that strain ADAl3–4 promotes plant growth via regulating signal transduction, phytohormone biosynthesis, flavonoid biosynthesis, and antioxidant capacity, etc., under aluminum stress. Our findings indicate that *Pseudomonas* sp. strain ADAl3–4 enhances plant development and stress resilience under Al^3+^ toxicity through a coordinated multi-dimensional regulatory network. Furthermore, strain ADAl3–4 promoted the root absorption of aluminum rather than the transportation of Al to the aerial part, endowing it with application prospects.

## 1. Introduction

Aluminum (Al) is the most abundant metallic element on Earth. Due to rapid industrialization, sewage irrigation, and the improper use of agrochemicals, aluminum accumulation in topsoil has intensified, making Al toxicity a critical limiting factor for crop production, particularly in acidic soils (pH < 5.0) where elemental Al exists as free aluminum ions (Al^3+^) [1,2,3,4,5]. These phytotoxic ions impair plant growth and biomass by inhibiting nutrient/water uptake and suppressing root development [1,3,4,5]. Furthermore, Al^3+^-induced alterations in soil physicochemical properties disrupt microbial community structure and diversity, compromising soil health. Approximately 50% of arable lands worldwide suffer from Al toxicity associated with acidic soils, establishing Al stress as a global agricultural constraint affecting plant growth and limiting crop productivity in acid-affected regions [2,6,7,8].

Previous studies demonstrate that Al toxicity predominantly damages root apical zones through multiple mechanisms: inducing DNA damage, suppressing metabolite biosynthesis, and triggering cell cycle arrest, thereby inhibiting both cell elongation and division [4,9,10]. These cellular disruptions result in stunted root development and impaired acquisition of essential nutrients including phosphorus, calcium, and magnesium [11,12,13]. In addition to root damage, Al toxicity causes systemic effects such as impaired stomatal regulation, decreased photosynthetic efficiency, chloroplast damage, reactive oxygen species (ROS) overproduction, and extensive peroxidation of membrane lipids [14]. The Al-induced ROS burst leads to lipid peroxidation, organelle dysfunction, and disruption of various metabolic pathways, resulting in widespread physiological imbalances [7,15,16].

Environmental aluminum (Al) circulates through food chains, posing significant threats not only to plants but also to animals and humans [17]. Excessive Al accumulation in edible plant tissues leads to diverse health complications, including musculoskeletal, hepatic, renal, neurological, and respiratory impairments in exposed organisms. Human exposure occurs primarily through drinking water and dietary intake, with multiple potential Al sources [18,19]. Notably, tea plants *Camellia sinensis* exhibit high Al concentrations in leaves, resulting in substantial Al content in tea infusions. Although gastrointestinal absorption of Al is limited and renal excretion proves relatively efficient, chronic exposure may induce severe health consequences or potential risks [20]. Accumulating evidence suggests associations between elevated Al body burdens and various disorders, including encephalopathy, dementia, Alzheimer’s disease, and osteomalacia-related fractures [18,19,20].

Current research has revealed that plants employ two principal strategies to counteract aluminum (Al) stress: blocking Al^3+^ entry and regulating Al tolerance mechanisms. The first strategy involves external exclusion mechanisms that prevent Al^3+^ infiltration into root tips, primarily through the chelation of Al^3+^ by root-exuded organic acids (OAs) such as citrate, malate, oxalate, and acetate, which form non-toxic, stable complexes with Al^3+^, effectively inhibiting their uptake into root cells [8,15,21,22,23]. Additional exclusion mechanisms include rhizosphere pH elevation, cell wall immobilization of Al^3+^, selective plasma membrane permeability, and active Al^3+^ efflux [9,24]. Modifications in cell wall composition play a pivotal role in Al resistance, as Al^3+^ interacts electrostatically with negatively charged pectin carboxyl groups [25] or adsorbs to uncharged hemicellulose polymers [26]. These interactions alter cell wall properties by reducing plasticity/elasticity, disrupting the activity of cell wall-modifying enzymes (e.g., expansins), and increasing soluble Al^3+^ accumulation in the apoplast [3,27]. The second strategy encompasses internal tolerance mechanisms activated post-Al^3+^ entry, including cytoplasmic chelation of Al^3+^ by OAs and subsequent sequestration of Al^3+^–OA complexes into vacuoles [8,15]. Vacuoles serve as primary storage sites for OAs, where malate and citrate synthesized in the cytoplasm are transported to enhance intracellular Al detoxification [28]. Furthermore, plants have evolved antioxidant defense systems to mitigate Al-induced oxidative stress, involving enzymatic components—superoxide dismutase (SOD), glutathione peroxidase (GPX), catalase (CAT), peroxidase (PRX), and ascorbate–glutathione reductase (AsA—GR) and non-enzymatic antioxidants (e.g., polyphenols, glutathione) [21,29,30]. Plants also modulate the expression of stress-responsive genes to reduce Al^3+^-mediated cellular damage [5,31]. Proteomic studies, particularly those utilizing KEGG pathway analysis, have further advanced the understanding of molecular mechanisms underlying Al toxicity tolerance in plants.

The plant rhizosphere microbiota constitutes a highly diverse microbial community inhabiting the soil region adjacent to plant roots. Plant growth-promoting rhizobacteria (PGPR) enhance plant growth by secreting beneficial compounds, modify plant heavy metal translocation pathways, improve stress resilience, and accelerate phytoremediation processes [32]. These rhizobacteria also regulate plant–environment interactions through diverse activities, playing critical roles in maintaining soil ecosystem health and functionality. For instance, co-inoculation with PGPR strains CABV2 and CASL5 promotes organic acid secretion, increases biomass production, and elevates antioxidant capacity in tomatoes to alleviate Al^3+^ stress [33]. Endophyte inoculation in tea plants *C. sinensis* enhances aluminum tolerance, as evidenced by reduced Al^3+^ accumulation and improved physiological–biochemical parameters [34]. Strain CAM4 mitigates Al toxicity by improving plant growth, elevating photosynthetic pigment content, decreasing proline and malondialdehyde (MDA) levels, enhancing antioxidant enzyme accumulation, and reducing Al uptake in host plants [35]. Strain CAH5, isolated from rhizospheric soil, significantly improves lettuce growth under combined Al and drought stress by attenuating oxidative stress, lipid peroxidation, and Al^3+^ accumulation in plant tissues, demonstrating its potential as a bioresource for Al-contaminated soil remediation [36]. Collectively, PGPRs modulate chemical secretion to ameliorate soil conditions, stimulate rhizosphere resistance-related protein synthesis, and reinforce plant stress adaptation mechanisms.

In this study, we isolated a plant growth-promoting rhizobacterium (PGPR), *Pseudomonas* sp. strain ADAl3–4, which significantly enhanced plant growth under aluminum stress. ADAl3–4 exhibits multiple growth-promoting traits, including IAA production, ACC deaminase activity, siderophore secretion, biofilm formation, phosphate solubilization, and nitrogen fixation. These activities modulate phytohormone homeostasis, photosynthesis, metabolism, and nutrient uptake. Interestingly, while ADAl3–4 increased aluminum uptake in roots, it did not significantly raise Al levels in shoots, highlighting its potential for phytostabilization and Al pollution remediation.

## 2. Results

### 2.1. Strain ADAl3–4 Promoted Plant Growth and Development Under Aluminum Stress

To elucidate the plant growth-promoting mechanisms by PGPR under aluminum (Al^3+^) stress, over 400 rhizospheric microbes were isolated from plants grown in Anda, Heilongjiang Province, China, using *Arabidopsis thaliana* (Col–0) as experimental material, and aluminum-supplemented medium (1/2 MS + 450 μM Al^3+^) to simulate Al stress. And strain ADAl3–4 was selected for its superior growth-promoting efficacy. Under Al stress, strain ADAl3–4-treated seedlings exhibited significantly longer primary roots and stronger shoots compared to the controls (Appendix A), demonstrating its ability to alleviate Al-induced growth inhibition. Soil cultivation experiments using alfalfa (*Medicago sativa*) and maize (*Zea mays*) (900 mg/kg Al for alfalfa; 1100 mg/kg Al for maize) revealed that strain ADAl3–4 application significantly restored root length, plant height, biomass (fresh/dry weight), and chlorophyll content while reducing proline, malondialdehyde (MDA) contents, and enhancing antioxidant enzyme activities (SOD, APX, POD, and CAT) in alfalfa (Figure 1A–D). Moreover, strain ADAl3–4 increased maize biomass, photosynthetic efficiency, antioxidant capacity, and osmotic adjust capacity while mitigating oxidative damage (Appendix A). These results indicate that strain ADAl3–4 enhances plant tolerance to Al stress by improving photosynthetic efficiency, adjusting ROS homeostasis and osmotic balance, thereby alleviating oxidative damage and growth inhibition, with broad-spectrum efficacy across species.

To analyze the soil remediation ability of strain ADAl3–4, key indicators of soil enzyme activity were detected, as they are significantly influenced by soil microbial communities [37]. Determination of the activities of urease and sucrase in rhizosphere soil of alfalfa showed that under normal condition, the application of strain ADAl3–4 significantly increased their activities. However, under Al stress, their activities were severely inhibited. After the inoculation of strain ADAl3–4, the activities of urease and sucrase were restored (Figure 1E), indicating that strain ADAl3–4 can restore the activity of Al^3+^-polluted soil. Furthermore, the Al^3+^ content in roots significantly increased after application of strain ADAl3–4, but did not have significant change in shoots (Figure 1F), suggesting that strain ADAl3–4 promotes both plant growth and Al^3+^ absorption, thereby reducing the Al^3+^ pollution in soil.

### 2.2. Genus Identification of Strain ADAl3–4 and Functional Characterization

To taxonomically classify strain ADAl3–4, a fragment of 16S rDNA gene was amplified and sequenced via Sanger sequencing. Phylogenetic analysis according to the NCBI database identified strain ADAl3–4 as *Pseudomonas* sp. (Figure 2A). To assess aluminum tolerance, *Pseudomonas* sp. strain ADAl3–4 was cultured in LB medium with a gradient of Al^3+^ concentrations. Compared with the Al^3+^-free control (OD_600_ = 1.86), strain ADAl3–4 maintained moderate growth, with OD_600_ values ranging from 1.69 to 1.32 under 250–1450 μM Al^3+^ treatments (Figure 2B), indicating a strong Al^3+^ tolerance. To investigate its plant growth-promoting potential, the secretory traits of strain ADAl3–4 were analyzed. Acidification assay showed that strain ADAl3–4 reduced the pH value of alkalinized LB medium (initial pH 8.0) to approximately 7.1 within 6 h post-inoculation (1:100 *v*/*v*, OD_600_ = 1.0), indicating the production of acidic compounds (Figure 2C). Further biochemical analyses revealed strain ADAl3–4’s abilities to produce 1–aminocyclopropane–1–carboxylate (ACC) deaminase, indole–3–acetic acid (IAA), siderophores, and biofilms (Figure 2D), along with phosphate-solubilizing (Figure 2E) and nitrogen-fixing (Figure 2F) capabilities. These results suggest that *Pseudomonas* sp. strain ADAl3–4 promotes plant growth through multiple synergistic mechanisms, including enhanced root colonization via biofilm formation, improved nutrient acquisition through phosphate solubilization, nitrogen fixation, and siderophore production, mitigation of ethylene-induced growth inhibition via ACC deaminase activity, and modulation of auxin levels.

### 2.3. Tanscriptomic Analysis of Alfalfa Roots Under Aluminum Stress with Application of Pseudomonas sp. Strain ADAl3–4

To elucidate the molecular mechanism of strain ADAl3–4 growth-promoting characteristics under aluminum stress, transcriptomic analysis was conducted using alfalfa roots treated with 900 mg/kg Al. Three experimental groups were established: H_2_O–CK (water-treated control), Al–CK (Al-treated control), and Al+ADAl3–4 (Al-treated with application of strain ADAl3–4). Principal component analysis (PCA) revealed significant divergence between Al–CK vs. H_2_O–CK and Al+ADAl3–4 vs. Al–CK groups (Figure 3A), identifying a total of 5580 differentially expressed genes (DEGs). DEGs were identified using a threshold of |log_2_FoldChange| ≥ 1 and FDR < 0.05. Specifically, the Al–CK vs. H_2_O–CK comparison exhibited 2335 upregulated and 2818 downregulated DEGs (Figure 3B), while the Al+ADAl3–4 vs. Al–CK comparison exhibited 441 upregulated and 390 downregulated DEGs (Figure 3C). Venn diagram analysis demonstrated 4749 unique DEGs for Al–CK vs. H_2_O–CK, 427 unique for Al+ADAl3–4 vs. Al–CK, and 404 shared between the two comparisons (Figure 3D and Appendix A). To demonstrate the molecular mechanism of alfalfa response to *Pseudomonas* sp. strain ADAl3–4 under aluminum stress, KEGG enrichment analysis was performed. The results indicated that in the Al–CK_vs_H_2_O–CK group, the DEGs were enriched in 139 pathways, while in the Al+ADAl3–4_vs_Al–CK group, the DEGs were enriched in 94 pathways. The TOP20 pathways of the two groups were mainly related to Biosynthesis of secondary metabolites, biosynthesis of amino acids, metabolic pathways, MAPK signaling pathway, flavonoid biosynthesis, plant hormone signal transduction, etc. (Figure 3E,F).

GO enrichment analysis indicated that in Al–CK_vs_H_2_O–CK group, the DEGs were classified into Biological process (BP), Cellular component (CC), and Molecular function (MF)—three categories including 32 subcategories (Appendix A). In CC (GO: 0005575), the DEGs were mainly enriched in Chloroplast nucleoid (GO: 0042644) and Chloroplast membrane (GO: 0031969) (Appendix A), etc. In MF (GO: 0003674), the DEGs were mainly enriched in Oxidoreductase activity, acting on the CH–CH group of donors, NAD or NADP as acceptor (GO: 0016628), Hydrolase activity, acting on ether bonds (GO: 0016801), Flavin adenine dinucleotide binding (GO: 0050660), and Monocarboxylic acid binding (GO: 0033293), etc. (Appendix A). In BP (GO: 0008150), the DEGs were mainly enriched in the Glutamine family amino acid metabolic process (GO: 0009064) and Oxylipin biosynthetic process (GO: 0031408), etc. (Appendix A). These results indicated the photosynthesis and redox processes were significantly affected under aluminum stress. Meanwhile, in Al+ADAl3–4_vs_Al–CK group, the DEGs were classified into the three categories including 30 subcategories (Appendix A). In BP, the DEGs were mainly enriched in the Reactive oxygen species metabolic process (GO: 0072593), Hydrogen peroxide metabolic process (GO: 0042744), Regulation of jasmonic acid-mediated signaling pathway (GO: 2000022), Negative regulation of defense response to insect (GO: 1900366), Response to wounding (GO: 0009611), etc. (Figure 4A). In MF, the DEGs were mainly enriched in jasmonic acid hydrolase (GO: 0120091), Protein phosphatase inhibitor activity (GO: 0004864), Isoprenoid binding (GO: 0019840), Alcohol binding (GO: 0043178) (Figure 4B). In CC, the DEGs were mainly enriched in Endoribon uclease complex (GO: 1902555), Cell wall (GO: 0005618), Secretory vesicle (GO: 0099503), and Glyoxysome (GO: 0009514), etc. (Figure 4C). These results showed that redox processes, hormone signal transduction, secretion processes, etc., were significantly affected after the application of *Pseudomonas* sp. strain ADAl3–4 under aluminum stress.

### 2.4. Pseudomonas sp. Starin ADAl3–4 Enhanced Flavonoid Biosynthesis Under Aluminum Stress

Plant flavonoids play vital roles in plant development and stress response. KEGG analysis indicated that Flavonoid biosynthesis (ko00941) was more significantly enriched in the Al+ADAl3–4 vs. Al–CK group than that in the Al–CK vs. H_2_O–CK comparison (Figure 5A,B). In the former comparison, multiple key genes involved in flavonoid biosynthesis, including *CHS*, *F3H*, *FLS*, *F3’H*, and *ANS*, exhibited significantly altered expression levels (Figure 5A). Chalcone synthase (CHS) catalyzes the condensation of malonyl–CoA and 4–coumaroyl–CoA to form naringenin chalcone, a critical initial intermediate in flavonoid biosynthesis. Chalcone subsequently serves as a precursor for the production of diverse flavonoid compounds. Hydroxycinnamoyl transferase (HCT) transfers hydroxycinnamoyl groups to specific acceptors, thereby contributing essential intermediates for flavonoid biosynthesis and influencing plant defense and stress adaptability. Flavonol synthases (FLSs) catalyze the conversion of dihydroflavonols to flavonols, potent antioxidants that protect cells from reactive oxygen species (ROS)-induced damage and enhance tolerance to both biotic and abiotic stresses. Anthocyanidin synthases (ANSs) catalyze the oxidation of colorless leucoanthocyanidins to form anthocyanins, which are involved in plant responses to environmental stimuli. RT-PCR analysis confirmed that under aluminum stress, the transcription levels of *CHS*, *FLS*, and two *HCTs* significantly increased after the application of strain ADAl3-4, which were 3.84, 1.35, 19.2, and 1.62 times higher than before treatment, respectively (Figure 5B). Consistently, total flavonoid content was also markedly elevated (Figure 5C), indicating that strain ADAl3–4 activated flavonoid biosynthesis pathways, which may enhance alfalfa tolerance to aluminum stress.

### 2.5. Pseudomonas sp. Strain ADAl3–4 Regulated Signaling Pathways Under Aluminum Stress

The signal transduction pathways of alfalfa respond to strain ADAl3–4 under aluminum stress included Plant hormone signal transduction and MAPK signaling pathway. There are some overlapping parts between the two pathways. For example, MAPK signaling pathway includes phytohormone, e.g., JA, ethylene, and ABA (Figure 6A). And the Plant hormone signal transduction pathway also includes these phytohormone pathways. In addition, it also includes auxin, cytokinin, gibberellin, etc. (Figure 6B). The expression levels of many key genes in these pathways altered significantly after inoculation of strain ADAl3–4 under aluminum stress, such as *FLS2, BAK1, PR1, JAZ, MYC2, SAUR, DELLA,* and some *WRKYs*, etc. The RT−qPCR validation of some of the genes were consistent with those of the transcriptome (Figure 6C). These results indicate that the strain ADAl3–4 activated the signaling pathways to enable plants to adapt to stress conditions. 

### 2.6. Pseudomonas sp. Strain ADAl3–4 Regulated Auxin Homeostasis and Cell Division Activity Under Aluminum Stress

Plant development is tightly regulated by phytohormones, particularly auxin, which orchestrates cell division, root/shoot organogenesis and development, and defense responses, etc. To decipher how strain ADAl3–4 modulates root development under aluminum stress, endogenous auxin dynamics were monitored using *Arabidopsis* DR5::GUS reporter lines, while mitotic activity was tracked via CYCB1;1::GUS lines, given CYCB1;1’s auxin-inducible expression and role in M-phase regulation. Histochemical assays demonstrated that strain ADAl3–4 application under non-stress conditions markedly elevated auxin accumulation in the quiescent center (QC) and columella cells (Figure 7A,B) and moderately enhanced mitotic activity in the root apical meristem (RAM) (Figure 7C,D). Under aluminum stress, auxin levels significantly increased, but mitotic activity was severely suppressed. Meanwhile, ADAl3–4 application further increased auxin level, restoring cell division activity (Figure 7A–D). These results revealed that strain ADAl3–4 promotes root development by recalibrating auxin gradients and activating cell cycle activity under aluminum stress.

To validate the above deduction, phenotypic assays were conducted using auxin transport mutants (*pin1, pin3, pin5, pin7*) and auxin biosynthesis mutants (*iaa17, ckrc1–1, yuc5, yuc8*). Under normal conditions, application of strain ADAl3–4 increased root length of wild type (Col–0) and the auxin-related mutants. However, the auxin transport mutants had lower root elongation ratio than that of wild type, except for *pin7*, which had a higher rate than that of wild type. However, the auxin synthesis mutants had higher root elongation ratio than that of wild type, except for *ckrc1–1*, which had a lower rate than that of wild type (Figure 7E–G). Under aluminum stress, root elongation of all genotypes were suppressed. After the application of strain ADAl3–4, their root elongation was restored. But only the root length recovery rate of *pin3* was close to that of the wild type, while *yuc5* recovered better than the wild type. The recovery of the remaining mutants was significantly lower than that of the wild type (Figure 7H–J). These results suggested that under both normal and aluminum stress conditions, the capacity of IAA secretion by *Pseudomonas* sp. strain ADAl3–4 may promote *Arabidopsis* development via compensating for the defects on auxin biosynthesis and polar transport in the mutants.

NRAMP1 is a plasma membrane-localized Mn^2+^ transporter [38,39]. The *nramp1* mutant was used to evaluate the impact of strain ADAl3–4 on ion transport. Under normal conditions, strain ADAl3–4 application increased *nramp1* root length by 1.89-fold. Under aluminum stress, *nramp1* root length did not change significantly, while the application of strain ADAl3–4 significantly increased *nramp1* root length (Figure 7E–J). This finding suggests that *Pseudomonas* sp. strain ADAl3–4 modulates ion homeostasis via an unknown mechanism under normal and Al stress conditions.

## 3. Discussion

Aluminum (Al) toxicity severely inhibits plant growth, and plants have evolved multiple resistance strategies to cope with Al stress. One well-known strategy is the exclusion mechanism, in which root tips secrete organic acid anions (e.g., malate, citrate) into the rhizosphere under Al stress. These anions chelate Al^3+^ ions in the apoplast, thereby preventing their entry into root cells and mitigating toxicity. This mechanism is distinct from internal tolerance strategies, which involve detoxification processes within cells after Al uptake. [40,41,42]. The organic acids malate, citrate, and oxalate are secreted by the roots and chelate Al^3+^ to form a non-toxic Al-OA complex, which decreases the entry of Al^3+^ into the root cells [35]. The *Pseudomonas* sp. (D95) and other strains contained in Synthetic Communities enhance the efficiency of monosaccharides in root exudates, including glucose, fructose, and furanose glucose, by promoting photosynthesis, providing carbon sources and energy for rhizosphere microbial communities. At the same time, Synthetic Communities’ phosphorus-solubilizing function has increased the nitrogen source for plants, thus significantly promoting rice growth [34]. The activation of flavonoid biosynthesis by strain ADAl3–4 in alfalfa contributed to Al detoxification probably through two main mechanisms: the chelation of Al^3+^ ions and scavenging of Al-induced reactive oxygen species (ROS) [43,44,45]. These functions are largely conferred by their adjacent phenolic hydroxyl groups, which enable the formation of stable Al–flavonoid complexes. Root-exuded flavonoids thus serve as effective extracellular ligands for Al neutralization [42,46,47]. These findings collectively indicate that PGPRs may alleviate Al toxicity through distinct yet complementary metabolic pathways, including flavonoid-mediated chelation, neutralization of protons, and ROS scavenging, etc. For instance, the highly Al-tolerant woody plant *Eucalyptus camaldulensis* secretes oenothein B via its roots, and this compound plays a critical role in aluminum tolerance [42]. Osawa et al. reported that proanthocyanidins (PAs) alleviate Al toxicity by forming stable complexes with Al^3+^ ions, thereby reducing Al uptake [40,41,42,46]. In addition, the strong antioxidant and free radical-scavenging activities of phenolic compounds help protect plant cells from Al-induced oxidative stress. Notably, Al exposure can further influence the antioxidant capacity and metal-chelating properties of polyphenols, suggesting a feedback mechanism in their response to Al stress [48,49].

Recent studies have highlighted the role of flavonoid biosynthesis in enhancing Al tolerance. For example, Su et al. revealed that the alfalfa transcription factor MsMYB741 enhances Al^3+^ resistance by promoting flavonoid accumulation under Al stress. In MsMYB741-overexpressing seedlings, key flavonoid biosynthetic genes (*F3H*, *FLS*, *PAL1*, *4CL*, *IFS*, *CHI*, and *CHS*) were significantly upregulated, accompanied by increased levels of 7,4′-dihydroxyflavone, liquiritigenin, and naringenin, underscoring the pivotal role of flavonoid biosynthesis in Al stress adaptation [50]. Recent research has also shown that *Bacillus altitudinis* AD13–4 enhances the accumulation of flavonoids and terpenoids under saline–alkaline stress, thereby promoting plant growth and biomass production [37]. Our study reveals for the first time that *Pseudomonas* sp. strain ADAl3–4 alleviates aluminum toxicity through the synergistic regulation of flavonoid and auxin pathways. We demonstrate that ADAl3–4 inoculation significantly promotes alfalfa growth under Al stress, coinciding with a marked increase in total flavonoid content in roots and dynamic upregulation of key flavonoid biosynthesis genes (Figure 5). This enhanced flavonoid biosynthesis likely facilitates the chelation and sequestration of Al^3+^ ions within the root system, limiting their translocation to shoots, as supported by the observed significant increase in root Al content without corresponding elevation in shoot Al levels (Figure 1F). This root-restricted aluminum accumulation mechanism effectively mitigates soil aluminum toxicity while preserving aerial plant parts, highlighting the potential application of ADAl3–4 in bioremediation of aluminum-contaminated soils.

Auxin, a key phytohormone, regulates both plant development and abiotic stress responses. Under saline–alkaline stress, *B. altitudinis* AD13–4 enhances plant growth by modulating auxin transport and mitotic activity, significantly increasing biomass [37]. Similarly, our study reveals that *Pseudomonas* sp. strain ADAl3–4 improves Al^3+^ tolerance and promotes growth through synergistic regulation of auxin homeostasis and flavonoid detoxification, while activating cell cycle activity. To further dissect the underlying mechanisms, we examined the effects of ADAl3–4 in *Arabidopsis* auxin transport and biosynthesis mutants. Under Al^3+^ stress, the auxin efflux transporter PIN1 emerged as a critical component in ADAl3–4-mediated adaptation, as the *pin1* mutant exhibited the most pronounced root shortening (Figure 7H). Interestingly, the *yuc5* mutant showed significant root elongation under Al stress, contrasting with other *yuc* mutants (Figure 7H), suggesting that YUC5 may act as a negative regulator in modulating ADAl3–4-mediated Al stress responses.

Collectively, these findings indicate that ADAl3–4 fine-tunes auxin homeostasis by influencing polar auxin transport and biosynthesis, with YUC5 serving as a critical node balancing growth and stress adaptation. However, the precise molecular mechanisms underlying this regulation warrant further investigation.

## 4. Materials and Methods

### 4.1. Plant Materials and Growing Conditions

*A. thaliana* wild-type (Col–0) and mutant seeds (*pin1* [SALK_097144C], *pin3* [SALK_113246C], *pin5* [SALK_042994C], *pin7* [SALK_048791C], *ckrc1–1* [SALK_127890C], *yuc5* [SALK_088618C], *yuc8* [SALK_096110C], *iaa17* [SALK_065697C], and *nramp1* [SALK_053236C]) [51,52,53,54,55,56,57,58] were surface-sterilized with 75% (*v*/*v*) ethanol for 5 min, rinsed three times with sterile distilled water (30 s each), and aseptically sown onto either basal medium (1/2 MS, pH 5.8) or aluminum stress medium (1/2 MS supplemented with 450 μM aluminum chloride [AlCl_3_], pH 5.8). Seeds were sown on the upper side of the medium while ADAl3–4 was inoculated at the opposite side. The seedlings were vertically cultivated at 22 °C, with a 16 h light/8 h dark photoperiod.

### 4.2. Strain Screening and Genus Identification

Rhizosphere soil samples (5 g, pH 5.4~6.2, from three sampling points spaced 600–1000 m apart) were collected from the roots of healthy wild plants growing in the suburban area of Harbin, Heilongjiang Province, China. This area was selected as a natural environment for isolating diverse rhizobacteria that may possess plant growth-promoting traits. The samples were suspended in 45 mL sterile distilled water, vortexed for 15 min, and allowed to settle for 10 min. A 1 mL aliquot of supernatant was mixed with 9 mL sterile water (designated as 10^−1^), and sequential dilutions were performed by transferring 1 mL from the previous dilution into 9 mL sterile water to generate successive gradients (designated as 10^−2^ to 10^−7^). Aliquots (0.1 mL) of each dilution were spread onto LB agar plates and incubated at 30 °C for 2–3 days. Individual colonies were isolated through serial streaking. Using *A. thaliana* as a model plant, 17 plant growth-promoting rhizobacteria (PGPR), including strain ADAl3–4, were screened for their ability to enhance aluminum stress tolerance under simulated conditions (1/2 MS medium supplemented with 450 mM Al^3+^, pH 5.8) [37].

The 16S rRNA gene of strain ADAl3–4 was amplified using specific primers, and the PCR products were sequenced via Sanger sequencing (Appendix A). A neighbor-joining phylogenetic tree was constructed via blasting in NCBI database (National Center for Biotechnology Information, rRNA/ITS databases) using MEGA X (version 10.2.6) software with 1000 bootstrap replicates to assess branch reliability [34,59].

### 4.3. Aluminum Tolerance and pH Adaptability of Strain ADAl3–4

To investigate the aluminum (Al) tolerance of strain ADAl3–4, a gradient Al stress assay was conducted in LB liquid medium containing Al^3+^ concentrations of 0, 250, 550, 850, 1150, and 1450 μM [60]. The logarithmic-phase culture (OD_600_ = 1.0) was inoculated into Al-supplemented medium at a 1:4 (*v*/*v*) ratio, followed by 24-h incubation at 30 °C with continuous shaking. Bacterial growth density (OD_600_) was monitored at 2-h intervals to construct real-time growth dynamics. Concurrently, proton secretion capacity was assessed using alkalinized LB medium (initial pH 8.0) under identical inoculation conditions, with pH variations measured at 1.5, 3, 4.5, and 6 h post-inoculation to analyze the strain’s microenvironment acidification regulatory capacity.

### 4.4. Characteristic Analysis of Strain ADAl3–4

In accordance with standardized protocols, the plant growth-promoting attributes of strain ADAl3–4 were comprehensively evaluated. Nitrogen fixation ability was assessed using nitrogen-free Ashby medium (KH_2_PO_4_ 0.2 g/L, NaCl 0.2 g/L, MgSO_4_·7H_2_O 0.2 g/L, K_2_SO_4_·2H_2_O 0.2 g/L, CaCO_3_ 5 g/L, glucose 5 g/L, mannitol 5 g/L; pH 7.0), where colony formation indicated diazotrophic potential. Siderophore production was evaluated in MKB medium using the Chrome Azurol S (CAS) assay. ACC deaminase activity was detected in DF minimal medium containing 3 mM 1–aminocyclopropane–1–carboxylic acid (ACC) as the sole nitrogen source. Phosphate solubilization capacity was quantified by molybdenum–antimony blue colorimetry following cultivation in medium with insoluble phosphate. Indole–3–acetic acid (IAA) production was determined using the Salkowski reagent assay. Biofilm formation was analyzed in 96-well plates by Giemsa staining, with absorbance measured at 570 nm. All assays were performed under controlled conditions, following established methodologies [61,62].

### 4.5. Soil Culture Experiment

High-temperature sterilized horticultural soil was used for pot cultivation, with 100 alfalfa seeds uniformly sown per pot (diameter: 25 cm, approximately 2.5 kg soil), in quadruplicate biological replicates. Plants were cultivated at a constant 25 °C under an 8 h light/16 h dark cycle, with daily monitoring of germination progress. Aluminum chloride (AlCl_3_) treatment (900 mg/kg) was initiated at the seedling stage (3 days post-germination), applied four times at 5-day intervals, while the H_2_O–CK control group received equivalent volumes of water. During the first treatment, 10 mL of ADAl3–4 bacterial suspension (OD_600_ = 1) was applied, with water substituted for the H_2_O–CK group. Leaf and root samples were collected on day 3 after the final treatment for subsequent analysis [63]. The maize aluminum stress pot experiment was conducted following a similar protocol as described in the referenced study, with optimized conditions [64]. Nine maize seeds were uniformly sown per pot (diameter: 10 cm, approximately 0.5 kg soil), and four biological replicates were performed. The plants were cultivated at a constant temperature of 25 °C under an 8 h light/16 h dark cycle, with daily monitoring of germination progress. Aluminum chloride (AlCl_3_) treatment at a concentration of 1100 mg/kg was initiated at the seedling stage (3 days post-germination), applied four times at 5–day intervals. The control group (H_2_O–CK) received equivalent volumes of water. During the first treatment, 10 mL of ADAl3–4 bacterial suspension (OD_600_ = 1) was applied, with water substituted for the H_2_O–CK group. Leaf and root samples were harvested on day 3 post-final treatment for subsequent analysis.

### 4.6. Determination of Physiological Data and Soil Enzyme Activity of Alfalfa

Fresh alfalfa leaves (0.5 g) were used for physiological assays. Chlorophyll content was determined by extracting pigments in 95% ethanol under dark conditions for 24 h, followed by absorbance readings at 665 nm and 649 nm using a UV–visible spectrophotometer [65]. Peroxidase (POD) activity was measured via the guaiacol oxidation method. The 4.0 mL reaction mixture contained 2.9 mL of 50 mM phosphate buffer (pH 7.0), 1.0 mL of 20 mM guaiacol, 1.0 mL of 20 mM H_2_O_2_, and 0.1 mL of enzyme extract. The increase in absorbance at 470 nm was monitored to evaluate enzyme activity. Superoxide dismutase (SOD) activity was assessed by its ability to inhibit the photochemical reduction of nitroblue tetrazolium (NBT). A total volume of 3.5 mL reaction mixture contained methionine, NBT, EDTA, riboflavin, and enzyme extract. Absorbance was measured at 560 nm after light exposure. Catalase (CAT) activity was quantified by tracking the decomposition of hydrogen peroxide, with absorbance reduction at 240 nm serving as the indicator of enzyme activity [66]. Ascorbate peroxidase (APX) activity was evaluated based on the decline in absorbance at 290 nm as ascorbic acid (AsA) was oxidized in the presence of H_2_O_2_ [66]. Leaf dry weight was determined after drying samples at 80 °C for 24 h to constant weight. Malondialdehyde (MDA) levels were determined via the thiobarbituric acid (TBA)–trichloroacetic acid (TCA) method. Samples (0.5 g) were homogenized in 10% TCA, centrifuged, and reacted with 0.6% TBA at 95 °C for 30 min. The absorbance of the supernatant was measured at 532 and 600 nm to calculate MDA concentration [67,68]. Proline content was quantified using acid ninhydrin colorimetry. Extracts reacted with acid ninhydrin reagent in a boiling water bath (100 °C) for 1 h, and after cooling and toluene extraction, the absorbance at 520 nm was recorded [67,68].

Rhizosphere soil enzyme activities were quantified following standard colorimetric protocols. Oven-dried (24 h) and sieved soil samples (used concurrently for 16S rRNA gene sequencing) were analyzed for sucrase and urease activities in triplicate. Sucrase activity was determined using the 3,5–dinitrosalicylic acid (DNS) method, with absorbance measured at 508 nm following incubation with sucrose substrate. Urease activity was assessed via sodium phenate–sodium hypochlorite colorimetry at 578 nm after incubation with urea substrate. Enzymatic activities were expressed as µg product released per gram of dry soil per hour [69].

### 4.7. Content of Total Flavone Compounds in Alfalfa Root

Total flavonoid content of alfalfa roots was determined using a colorimetric method. Dried and sieved root powder (0.5 g) was extracted with 25 mL of 80% ethanol (1:50, *w*/*v*) via ultrasonication (320 W, 70 °C, 60 min). The extract was centrifuged at 10,000× *g* for 15 min, and 1 mL of the resulting supernatant was mixed sequentially with 0.5 mL of 5% NaNO_2_ (6 min), 0.5 mL of 10% Al(NO_3_)_3_ (6 min), and 4 mL of 4% NaOH. The mixture was brought to 10 mL with 70% ethanol and incubated for 15 min in the dark. Absorbance was measured at 510 nm using a UV–Vis spectrophotometer. All measurements were performed in triplicate [66,70].

### 4.8. Alfalfa RNA Isolation, Library Construction, RNA Sequencing, RT–qPCR

Root tissues used for RNA sequencing were identical to those employed in physiological assays. Total RNA was isolated using TRIzol reagent (Invitrogen, Carlsbad, CA, USA) following the manufacturer’s protocol. Library preparation, RNA sequencing, and RT–qPCR workflows were conducted as previously described [71]. For RT–qPCR, three independent biological replicates per sample were analyzed with triplicate technical repetitions. ACTIN was used as the internal reference gene, and all primers, including ACTIN, were confirmed to yield specific amplification products through PCR and melt curve analysis. Primer sequences are provided in Appendix A. Statistical analyses were performed using Student’s *t*-test with a significance threshold * *p* < 0.05; ** *p* < 0.01; *** *p* < 0.001. Raw transcriptomic reads were deposited in the NCBI Sequence Read Archive (SRA) database (Accession number: PRJNA1253887).

### 4.9. Transcriptome Sequencing and Analysis

Alfalfa samples were subjected to total RNA extraction using CTAB–PBIOZOL reagent, followed by quality verification via NanoDrop spectrophotometry and Agilent 2100 Bioanalyzer. Reverse transcription synthesized cDNA for PCR amplification, with amplicons purified by Ampure XP magnetic beads, eluted in EB buffer, and processed through dsDNA denaturation and oligonucleotide-mediated circularization to construct single-stranded circular DNA (ssCir DNA) libraries. Paired-end sequencing was performed on the Illumina HiSeq X Ten platform, and raw reads were filtered by fastp to remove adapter sequences and low-quality reads. High-quality reads were aligned to the alfalfa reference genome (Accession number: GCA_003473485.1) using HISAT2, with FPKM values calculated based on gene length and mapped read counts. DEGs were identified using DESeq2 (v1.22.1) with thresholds|log_2_(Fold Change)|≥ 1 and FDR < 0.05, adjusted via the Benjamini–Hochberg method [71]. KEGG pathway and GO term enrichment analyses were conducted using hypergeometric tests: KEGG focused on metabolic pathway units, while GO analysis adhered to term hierarchy [72].

## 5. Conclusions

This study reveals the molecular basis of *Pseudomonas* sp. strain ADAl3-4 strain alleviating the inhibition of plant growth by Al^3+^ stress through multiple synergistic mechanisms. Strain ADAl3-4 enhanced plant antioxidant capacity, regulated endogenous hormone and ion homeostasis, reprogramed the cell cycle of meristematic tissues to improve cell division activity via regulating plant secondary metabolism (such as Flavonoid biosynthesis pathway), plant signaling pathways (such as Plant hormone signaling pathway and MAPK signaling pathway), etc. These mechanisms synergistically promote the growth and development of various plants under Al^3+^ stress, increasing biomass and stress resistance. Meanwhile, strain ADAl3-4 also increased the absorption of Al^3+^ by roots, but restricted the transport of Al^3+^ to the aerial parts. These findings highlight the broad-spectrum and efficient promotion of plant growth by *Pseudomonas* sp. strain ADAl3-4 through the integration of a multi-dimensional regulatory network of “Stress–Signaling –Metabolism–Development”, while reducing soil Al pollution.

## Figures and Tables

**Figure 1 ijms-26-04919-f001:**
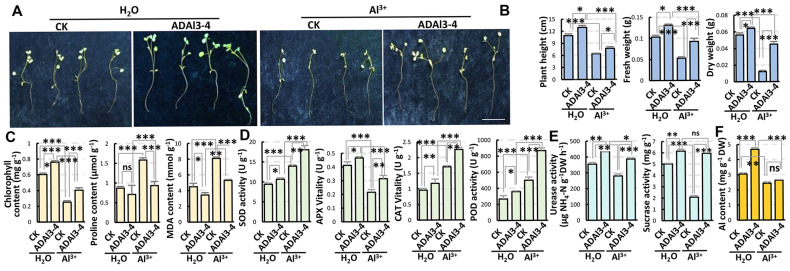
Strain ADAl3−4 promotes growth and alleviates Al^3+^ toxicity in alfalfa seedlings. (**A**) Representative phenotype of alfalfa (*M. sativa*) seedlings grown for 1 month in Al^3+^-contaminated soil (900 mg/kg Al^3+^) with or without ADAl3−4 inoculation. Scale bar = 5 cm. (**B**) Bar chart showing plant height, fresh weight, and dry weight of alfalfa under different treatments. (**C**) Bar chart showing chlorophyll, proline, and malondialdehyde (MDA) contents in alfalfa. (**D**) Activities of antioxidant enzymes in alfalfa roots, including superoxide dismutase (SOD), ascorbate peroxidase (APX), catalase (CAT), and peroxidase (POD). (**E**) Bar graph showing urease and sucrase activities in rhizosphere soil of alfalfa under normal and Al^3+^ stress conditions, with or without inoculation of strain ADAl3−4. (**F**) Statistics of Al content in roots and shoots of alfalfa seedlings. For panels (**B**−**F**), all data are mean ± SE of three biological replicates. Statistical analysis was performed using Student’s *t*-test. *: *p* < 0.05; **: *p* < 0.01; ***: *p* < 0.001; ns, no significance.

**Figure 2 ijms-26-04919-f002:**
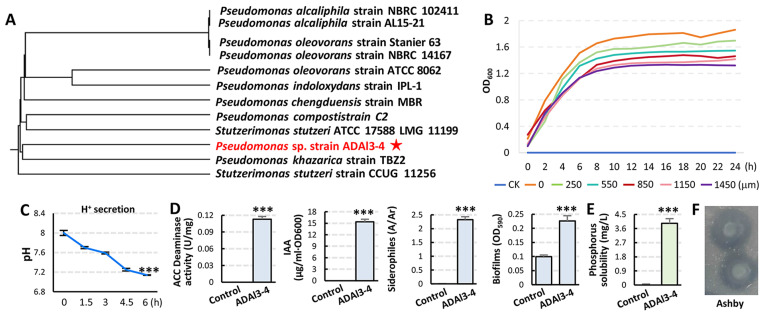
Phylogenetic analysis and functional characterization of *Pseudomonas* sp. strain ADAl3–4. (**A**) Phylogenetic tree constructed using neighbor-joining (NJ) method based on 16S rDNA sequences of indicated *Pseudomonas* strains. Red star, ADAl3–4. (**B**) Growth curves of *Pseudomonas* sp. strain ADAl3–4 under gradient of aluminum concentrations (0–1450 μM Al^3+^). (**C**) Acidification assay of bacteria culture medium. Dynamic pH values (initial pH 8.0) were recorded every 1.5 h within 6 h post-inoculation; three parallel experiments were conducted. (**D**,**E**) Quantitative evaluation of plant growth-promoting traits of *Pseudomonas* sp. strain ADAl3−4, including ACC deaminase activity, indole–3–acetic acid (IAA) production, siderophore production, biofilm formation, and phosphate solubilization capacity. All data are mean ± SE of three biological replicates per sample. *** *p* < 0.001. Student’s *t*-test. (**F**) Nitrogen fixation capability was assessed using Ashby nitrogen-free medium.

**Figure 3 ijms-26-04919-f003:**
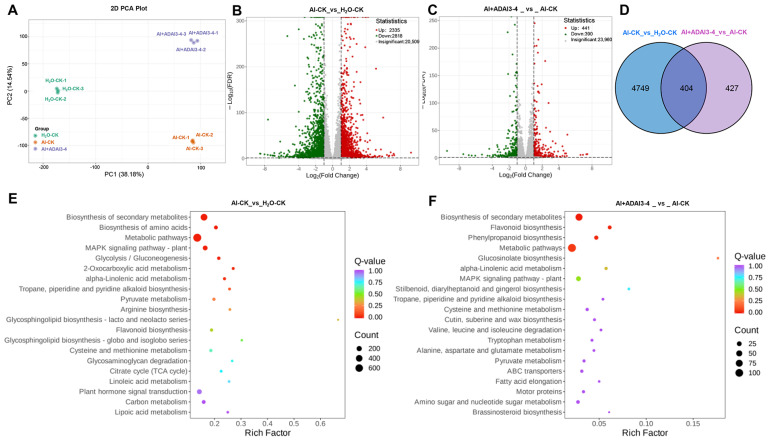
Transcriptome analysis of alfalfa roots under aluminum stress with the application of *Pseudomonas* sp. strain ADAl3−4. (**A**). The principal component analysis (PCA) showed differences in gene expression profiles between the indicated groups. (**B**,**C**). The volcano plots of the indicated groups. (**D**). The Venn diagram analysis of the specific and shared DEGs of the indicated groups. (**E**,**F**) KEGG enrichment (TOP 20) analysis of Al−CK vs. H_2_O−CK (**E**) and Al+ADAl3−4_vs_Al−CK (**F**) comparisons. The X−axis represents the enrichment factor; the Y-axis denotes pathways. The color gradient indicates Q−values; the circle size indicates the DEG number.

**Figure 4 ijms-26-04919-f004:**
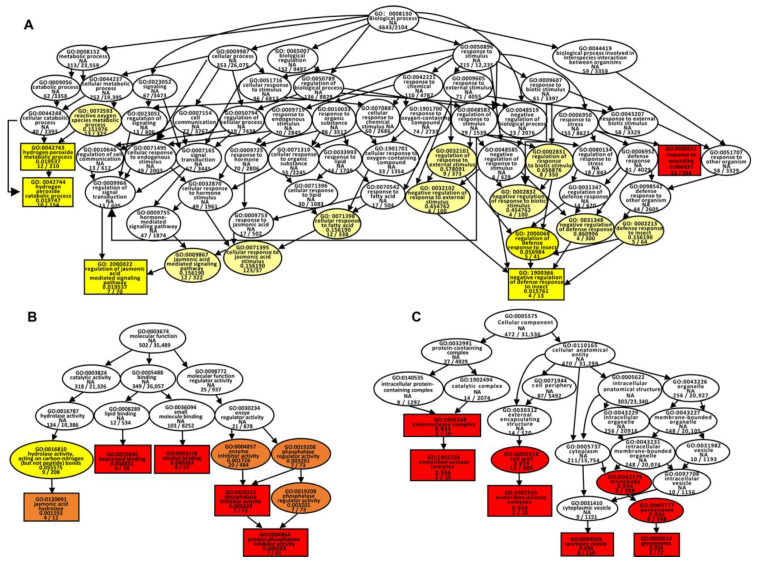
Gene Ontology (GO) enrichment analysis of DEGs in the Al+ADAl3–4_vs_Al–CK comparison. GO enrichment of differentially expressed genes (DEGs) in (**A**) biological processes, (**B**) molecular functions, and (**C**) cellular components. The color intensity represents statistical significance: red (*p* < 0.0001), orange (*p* < 0.001), light yellow (*p* < 0.05), and white (no significance).

**Figure 5 ijms-26-04919-f005:**
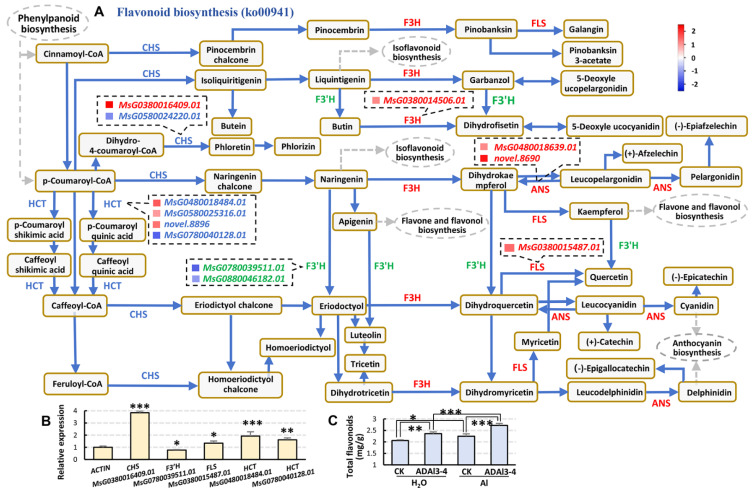
The activation of the flavonoid biosynthesis pathway by *Pseudomonas* sp. strain ADAl3–4. (**A**) The differential expression of key genes in the flavonoid biosynthesis pathway in the Al+ADAl3–4 vs. Al–CK group. The gene names in red, upregulated genes; in green, downregulated genes; in blue, both up- and downregulated genes in the same family. The blue and red blocks in front of the gene numbers represent gene expression levels according to the color scale. The gray dotted lines and rectangles indicate downstream pathways. The same apply in Figure 6A,B. (**B**) RT–qPCR validation of selected genes under aluminum stress following ADAl3–4 inoculation. All data are the mean ± SE of three biological replicates, with three technical replicates per biological replicate. *ACTIN* was used as endogenous reference. (**C**) The quantification of total flavonoid content in alfalfa roots under different treatments. All data are mean ± SE of three biological replicates. Statistical significance was assessed using Student’s *t*-test. *, *p* < 0.05; **, *p* < 0.01, ***, *p* < 0.001. Abbreviations: CHS, chalcone synthase; HCT, hydroxycinnamoyl transferase; F3H, flavanone 3–hydroxylase; F3’H, flavonoid 3′–hydroxylase; FLS, flavonol synthase; ANS, anthocyanidin synthase.

**Figure 6 ijms-26-04919-f006:**
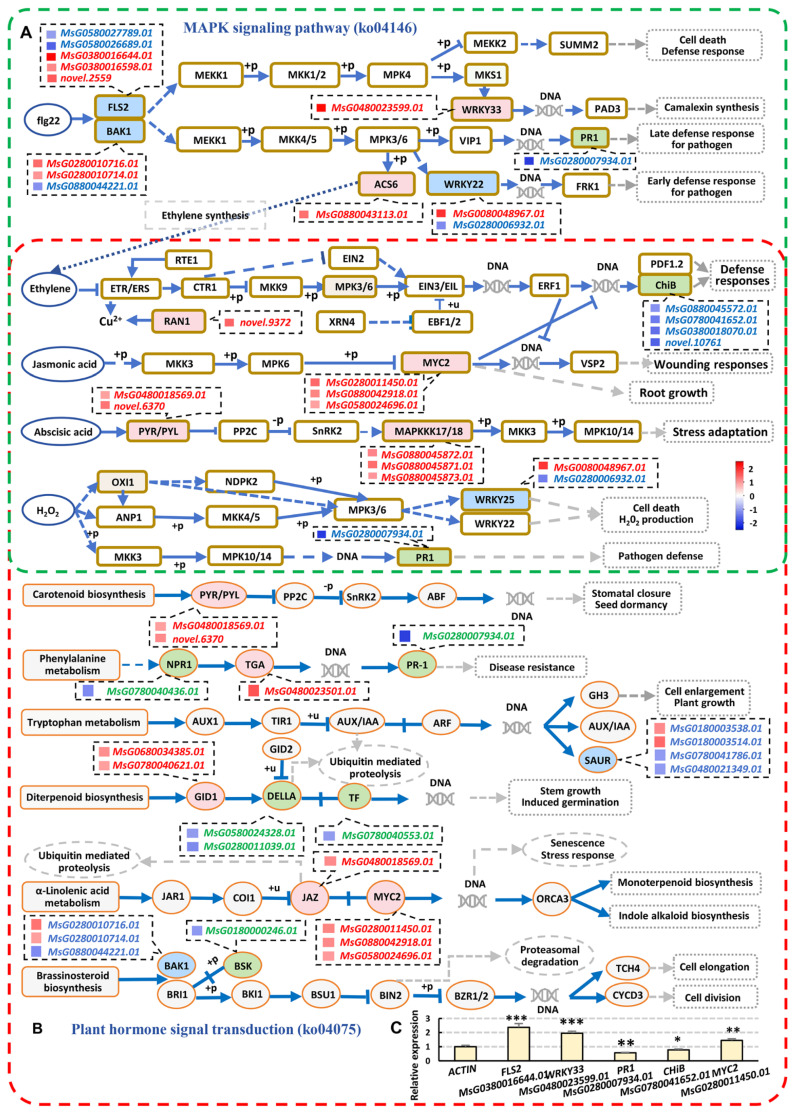
Signal transduction pathways for the Al+ADAl3–4 vs. Al–CK comparison. (**A**,**B**) Differentially expressed genes (DEGs) enriched in the MAPK signaling pathway (ko04146, (**A**)) and plant hormone signal transduction pathway (ko04075, (**B**)). The circles and frames with a red background, upregulated genes; with a green background, downregulated genes; and with a blue background, both up- and downregulated genes in the same family. The overlapping part of the two pathways are enclosed by both red and green frames. (**C**) RT–qPCR validation of the indicated genes. There are three technical replicates per experiment, and three independent experiments per sample. * *p* < 0.05; ** *p* < 0.01; *** *p* < 0.001. Student’s *t*-test. Abbreviations: FLS2, Flagellin Sensitive 2; BAK1, BRI1-Associated receptor Kinase 1; PR1, pathogenesis-related protein1; ACS6, 1–aminocyclopropane–1–carboxylate synthase 6; RNA1, RAS-related nuclear protein 1; ChlB, Calmodulin-binding protein B; MYC2, MYC transcription factor 2; PYR/PYL, Pyrabactin Resistance 1/PYR1-like receptors; NPR1, Nonexpressor of Pathogenesis-Related Genes 1; TGA, TGACG motif-binding protein; SAUR, Small Auxin–Up RNA; GID1, Gibberellin-Responsive D1 protein; JAZ, Jasmonate ZIM-domain protein; DELLA, DELLA domain protein; BSK, Brassinosteroid Signaling Kinase; BIN2, Brassinosteroid Insensitive 2.

**Figure 7 ijms-26-04919-f007:**
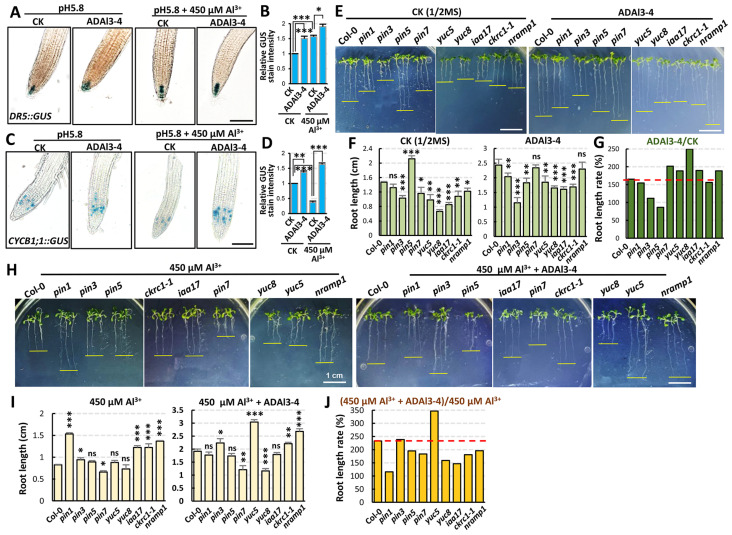
*Pseudomonas* sp. strain ADAl3–4 regulates root development in *Arabidopsis* by modulating auxin homeostasis under aluminum stress. (**A**) DR5::GUS reporter gene expression showing auxin distribution in *Arabidopsis* root tips under different treatments. (**B**) Quantification of GUS signal intensity in root tips based on image analysis. (**C**) CYCB1;1::GUS reporter gene expression indicating changes in cell division activity in the root apical meristem under various treatments. (**D**) Quantification of CYCB1;1::GUS signal intensity in root tips. (**E**,**F**) Root length variation in wild type and auxin-related mutants upon ADAl3–4 inoculation under normal conditions. The included mutants involve auxin transport (pin1, pin3, pin5, pin7), auxin biosynthesis and signaling (iaa17, ckrc1-1, yuc5, yuc8), and aluminum ion transport (nramp1). (**E**) Root length images of wild type and mutants (pin1, pin3, pin5, pin7, iaa17, ckrc1-1, yuc5, yuc8, and nramp1) with or without ADAl3–4 inoculation on ½ MS medium. (**F**) Quantification of root length from wild type and mutants with or without ADAl3–4 inoculation. (**G**) Comparison of root length between wild type and mutants upon ADAl3–4 inoculation. (**H**,**I**) Root length variation in wild type and the same set of mutants under 450 μM Al^3+^ stress. (**H**) Root length images of wild type and mutants (pin1, pin3, pin5, pin7, iaa17, ckrc1-1, yuc5, yuc8, and nramp1) with or without ADAl3–4 inoculation on 1/2 MS medium containing 450 μM Al^3+^. (**I**) Quantification of root length from wild type and mutants with or without ADAl3–4 inoculation under Al^3+^ stress. (**J**) Comparison of root length between wild type and mutants upon ADAl3–4 inoculation under aluminum stress. Scale bar = 1 cm; n = 30 per treatment. All data are mean ± SE of three independent experiments per sample. Statistical analysis was performed using Student’s *t*-test. *, *p* < 0.05; **, *p* < 0.01; ***, *p* < 0.001; ns, no significance.

## Data Availability

Data are contained within the article and Appendix A.

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
