# Peer review of "Pseudomonas sp. Strain ADAl3–4 Enhances Aluminum Tolerance in Alfalfa (Medicago sativa)"

_ijms, 2025, doi:10.3390/ijms26104919_

Round 1
Reviewer 1 Report
Comments and Suggestions for Authors
This study systematically elucidated the mechanisms by which the rhizobacterium Pseudomonas sp. ADAl3-4 promotes the growth of alfalfa (and Arabidopsis, maize) under Al³⁺ stress through multiple approaches including physiological, biochemical, transcriptomic, and Arabidopsis mutant validation, focusing on three synergistic pathways: bacterial induction of flavonoid biosynthesis, regulation of hormone (especially auxin) signaling and cell division activity, and NRAMP1-mediated ion balance. The work possesses certain innovation and application prospects, but there are still several aspects that need enhancement and clarification.
- Solely relying on 16S rRNA sequences for identification is inadequate for accurately determining the strain's species. It is advisable to complement this with whole-genome sequencing.
- Some figures have text that is too small and have low resolution. It is recommended to enhance the image clarity.
- The formatting of some references is inconsistent, please revise them carefully in accordance with the formatting requirements of journal. (“Medicago Truncatula” in line 594 should be italicized).
- The manuscript contains a few grammatical and spelling errors (For example, “Camellia sinensis” in line 59 should be italicized; the ‘n’ in nramp is not italicized in line 338; ‘Aluminum stress’ and ‘Al stress’ are repeated multiple times), and it is recommended to have it proofread by a native editor or a professional proofreading company.
Reviewer 2 Report
Comments and Suggestions for Authors
Why the specified soil sample was selected (line 417). Can You explain?
Reviewer 3 Report
Comments and Suggestions for Authors
The authors conducted a series of experiments demonstrating the role of Pseudomonas sp. ADAl3-4 in modulating plant auxin signaling. Furthermore, they provided evidence—through genetic analyses and RNA-seq—that this strain enhances plant tolerance to aluminum-induced stress. However, the manuscript contains numerous critical flaws, making it difficult to provide a proper scientific review.
Most notably, the manuscript frequently fails to employ appropriate model plant mutants as genetic tools, which significantly weakens the experimental rigor. In addition, Pseudomonas sp. is not consistently treated as an ecotype or strain throughout the manuscript, leading to ambiguity. The Materials and Methods section lacks citations for key protocols and provides only superficial descriptions of experimental procedures, which undermines reproducibility. The use of the term "AlCl₃" is also inappropriate and does not align with standard terminology in plant stress physiology. Moreover, the figure legends are insufficiently detailed and require substantial elaboration.
Overall, the current submission is not in a state suitable for peer review. A thorough revision and restructuring of the manuscript is required before it can be meaningfully evaluated.
